# Effects of Limestone Powder on the Early Hydration Behavior of Ye’elimite: Experimental Research and Thermodynamic Modelling

**DOI:** 10.3390/ma15196645

**Published:** 2022-09-25

**Authors:** Jian Ma, Ting Wang, Hu Shi, Zhuqing Yu, Xiaodong Shen

**Affiliations:** 1College of Materials Science and Engineering, Nanjing Tech University, Nanjing 211816, China; 2The Synergetic Innovation Center for Advanced Materials, Nanjing 211816, China

**Keywords:** hydration, limestone powder content, gypsum content, thermodynamic modelling, chemical shrinkage

## Abstract

To investigate the effects of limestone powder and gypsum on the early hydration of ye’elimite, the hydration behavior of C_4_A_3_$\bar{S}$-LP-CaSO_4_·2H_2_O-H_2_O systems are researched. The hydration behavior of systems are researched by employing isothermal calorimetry, XRD technique and chemical shrinkage. The thermodynamic modelling method is employed to predict the equilibrium phase assemblages. The results show that the system with 5 wt.% LP has a comparable hydration heat evolution to limestone powder-free systems. Limestone powder can take part in the reaction to produce monocarboaluminate in the system with M-value (molar ratio of gypsum to ye’elimite) of 1, but monocarboaluminate is not found in the system with M-value of 2. The level off time of chemical shrinkage shortens with the increase of limestone powder dosage. Thermodynamic modelling results show that monocarboaluminate is no longer formed in all systems when M-value exceeds 1.27, which corresponds to the XRD results. This study can provide theoretical guidance for the rational utilization of limestone powder in calcium sulphoaluminate cement.

## 1. Introduction

The serious consequences of climate change caused by the greenhouse effect can be mitigated by the carbon neutrality policy, which is a global mission [1]. The cement industry currently accounts for 5–8% of global carbon dioxide production [2]. The development and utilization of a new cementitious binder is an effective method to decrease the carbon footprint of the cement industry [3,4]. At present, non-tradition cementitious materials have received extensive attention, such as calcium sulphoaluminate (CSA) cement, alkali-activated materials [5,6,7,8], etc. The carbon emissions from manufacturing CSA cement are about 25% less than those from producing traditional Portland cement [9]. Besides, the advantages of CSA cement are low alkalinity, high early strength [10] and excellent permeability resistance [11,12].

Ye’elimite (C_4_A_3_$\bar{S}$) is the primary mineral phase in CSA cement, which is about 50–80 wt.% of CSA cement [13]. The crystal structure of C_4_A_3_$\bar{S}$ is porous, and mainly consists of Al-O tetrahedral group [14,15]. The unique crystal structure makes ye’elimite have high hydration activity. The hydration reaction rate of C_4_A_3_$\bar{S}$ is fast, resulting in the quick setting of CSA cement. Therefore, gypsum or anhydrite is usually present in the CSA cement to adjust the hydration reaction of cement [16]. The reaction of ye’elimite relies on the relative molar ratio of gypsum to ye’elimite, viz. the M-value [17]. For instance, ye’elimite reacts with water to produce monosulphoaluminate (AFm) and aluminium hydroxide (AH_3_) when the M-value is 0. When the M-value is 2, ye’elimite reacts with gypsum to form ettringite and AH_3_ [17]. The reaction products of CSA cement are directly related to M-value, which also affect the properties of CSA cement, such as compressive strength [18], carbonization resistance [19], expansion [20], etc. 

In addition to being a raw material to generate Portland cement, limestone powder (LP) can also be used to partially replace cement to reduce the carbon emission. In recent years, the use of LP in CSA cement binders has attracted much attention from researchers. The early hydration and compressive strength of CSA cement can be improved by incorporation of LP [21,22]. In addition, the reaction between LP and ye’elimite can generate hemicarboaluminate (Hc) and monocarboaluminate (Mc) [23,24]. The generation of Hc or Mc can stabilize ettringite, leading to the reduction of porosity of CSA cement at an early age [25]. The reaction of LP with ye’elimite is the crucial factor influencing the properties and hydration behavior of CSA cement binders. Besides ye’elimite, however, there are some other mineral phases in CSA cement, such as dicalcium silicate (C_2_S), tricalcium aluminate (C_3_A), ferrite (C_4_AF), etc. In addition, LP can also react with C_3_A to generate calcium carboaluminate [26,27,28]. Meanwhile, the M-value is the vital factor affecting the reaction of LP in the CSA binder. To investigate the role of LP in the CSA and simplify the system, it is therefore first worth studying the reaction between LP and ye’elimite thoroughly. 

Considering the crucial factor, M-value, the early hydration behavior of C_4_A_3_$\bar{S}$-LP-CaSO_4_·2H_2_O-H_2_O system is studied comprehensively. The used LP dosages are 0, 5, 15, and 25 wt. %, and the M-values employed in the study are 1 and 2. Some experiments, such as X-ray diffraction (XRD), isothermal calorimetry and chemical shrinkage (CS), are utilized to explore the hydration process of the whole system. In the end, the equilibrium phase assemblages of the system are predicted by the thermodynamic modelling method.

## 2. Materials and Methods

### 2.1. Materials

The synthetic ye’elimite was used in this research. It was prepared by analytical-grade reagents, CaCO_3_, Al_2_O_3_ and CaSO_4_·2H_2_O, in order to ensure the purity of ye’elimite. The molar ratio of CaCO_3_, Al_2_O_3_ and CaSO_4_·2H_2_O is 3:3:1. Dry powder materials were mixed for 3 h in a blender mixer, and then compacted by a compression machine. After preprocessing, ye’elimite was synthesized by sintering compacted powder at 1350 °C for 2 h (see Figure 1b). The synthetic ye’elimite was handled by a vibration mill and passed through a sieve with 75 μm mesh. XRD pattern of synthetic ye’elimite phase is shown in Figure 1a. The ye’elimite, mayenite and calcium aluminate are the primary mineral composition of synthetic ye’elimite. Limestone rock was ground by a laboratory ball mill to produce LP. Table 1 shows the chemical compositions of LP. The main mineral composition of LP is calcite (Figure 1c). The gypsum used in the experiments is an analytical grade reagent. 

### 2.2. Experimental Procedure

Different M-value and LP content were investigated in this research. Table 2 shows the mixture compositions used. Each sample is named YG(x)L(y), the M-value is denoted by x and the content of LP is denoted by y. Before casting, all materials were mixed by a blender mixer at the speed of 500 rpm for 30 min. Then, the mixture was blended with water by using a mixer for 2 min at a speed of 600 rpm. The ratio between water and binder is set as 1. Afterwards, the pastes were cast into a centrifuge tube with a volume of 10 mL. All samples were sealed cured at 20 ± 1 °C. At 1 h, 3 h, 6 h, 12 h, 1 d, 3 d and 7 d, and the specimen was soaked into ethanol in order to terminate the reaction of ye’elimite. 

### 2.3. Experimental Test

#### 2.3.1. Isothermal Calorimetry

Hydration heat evolution of all specimens were determined by an isothermal calorimeter at 20 °C. 4 g of powder and 4 g of water were intermixed for 2 min to prepare test samples. Then, the sealed ampoules were loaded into the isothermal calorimeter. The hydration heat data was recorded within 48 h.

#### 2.3.2. X-ray Diffraction

Mineral composition of all samples was detected by an X-ray diffractometer (CuKα, λ = 1.54 Å). The samples for the XRD test were ground with ethanol solution and dried in a vacuum drying oven at 40 °C for 24 h. The characteristic diffraction peaks of the main hydration products of C_4_A_3_$\bar{S}$-LP-CaSO_4_·2H_2_O-H_2_O system mainly appears in the range of 5–30° in the XRD pattern, and therefore the data of 2θ angles between 5° and 30° were collected, and the scan speed is 5 °/min.

#### 2.3.3. Chemical Shrinkage Test

The CS of system was detected in the light of standard ASTM C1608-2017. The raw materials were mixed with water by an overhead stirrer at a speed of 600 r/min for 2 min. Then, the mixtures were put in a glass vial (φ 22 × 55 mm) and tapped on the worktable. Deionized water was poured into the glass vial until it was full. Subsequently, a rubber stopper with capillary tube (capacity of 1.0 mL, accuracy of 0.01 mL) was inserted into the glass vial. Deionized water was poured into a capillary tube until to the top mark of the graduations. The oil was dripped onto the deionized water to avoid the evaporation of water. The test temperature was 20 ± 1 °C.

Data were recorded every 1 h until 8 h. After 8 h, the data were recorded every 8 h. The chemical shrinkage is calculated by Equations (1) and (2).
(1)$$M_{paste}=\frac{M_{\left(vial+paste\right)}-M_{vial}}{1.0+w/c}$$
(2)$$CS(t)=\frac{h(t)-h(60min)}{M_{cement}}$$

*M_cement_*: mass of paste in the vial, *g*; *M_(vial + paste)_*: mass of vial and cement paste, *g*; *M_vial_:* mass of empty vial, *g*; *CS(t)*: chemical shrinkage at time t, *mL/g*; *h(t):* water level in capillary tube at time t, *mL*; and *h(60 min)*: water level in capillary at *60 min*, *mL*.

#### 2.3.4. Thermodynamic Modelling Analysis

Phase assemblages at equilibrium condition of system can be predicted by thermodynamic methods. Gibbs energy minimization software GEMS3 was employed to perform thermodynamic modelling [29,30]. Thermodynamic data was obtained from the GEMS-PSI thermodynamic database [31] and CEMDATA database [32,33,34,35]. The activity coefficients of aqueous species were calculated by extended Debye-Hückel equation with the common ion size parameter a_i_ of 3.67 Å for KOH [36]. In the modelling process, the limestone powder was recognized as pure calcite. The calculating temperature was set as 20 °C.

## 3. Results and Discussions

### 3.1. Hydration Kinetics

Figure 2 presents the heat curves of the C_4_A_3_$\bar{S}$-LP-CaSO_4_·2H_2_O-H_2_O system when the M-value is 1. As shown in Figure 2a, after the induction period, each normalized heat flow curve has two exothermic peaks. The 1st exothermic peak is mainly related to the formation of ettringite. It is obvious that the incorporation of 25 wt.% LP retards the occurrence time of the 1st peak. In other words, more LP can retard the formation of ettringite. Around 4 h, the 2nd exothermic peak appears, which is primarily due to the formation of the AFm phase. It can be seen clearly that the LP retards the generation of AFm phase regardless of the LP dosage. The cumulative heat curves of the C_4_A_3_$\bar{S}$-LP-CaSO_4_·2H_2_O-H_2_O system (M-value = 1) within 48 h are shown in Figure 2b. The total heat decreases with the increase of the LP dosage, and YG1L5 has a comparable hydration heat with YG1L0. The total hydration heat is decreased by 20.6% when the dosage of LP comes up to 25 wt.% for 48 h, which is attributed to the dilution effect of LP.

Increasing the M-value from 1 to 2, viz. increasing the amount of gypsum, the hydration heat of the C_4_A_3_$\bar{S}$-LP-CaSO_4_·2H_2_O-H_2_O system is presented in Figure 3. As shown in Figure 3a, there is only one exothermic peak after the induction period for all samples. The first exothermic peak is also related to the formation of ettringite. The incorporation of 5 wt.% LP has little effect on the occurrence time of the 1st peak. However, the occurrence time of 1st exothermic peak is obviously retarded when the dosage of LP is 15 wt.% and 25 wt.%. Compared to the C_4_A_3_$\bar{S}$-LP-CaSO_4_·2H_2_O-H_2_O system (M-value = 1), there is no appearance of 2nd peak, viz. the formation of AFm. In theory, C_4_A_3_$\bar{S}$; can fully react with gypsum to form ettringite when the M-value is 2. As shown in Figure 3b, the total heat decreases after addition of LP due to the dilution effect, especially for the samples of YG2L15 and YG2L25.

Based on Figure 2b and Figure 3b, the increase of gypsum content obviously results in an increase of the cumulative heat of the system. When the M-value is 1, the hydration heat mainly comes from the reaction between gypsum and ye’elimite and the reaction between ye’elimite and water. When the M-value is 2, the hydration heat is only generated by the hydration reaction between ye’elimite and gypsum. The hydration heat generated by the reaction of ye’elimite with water is lower than that generated by the reaction of ye’elimite with gypsum. Therefore, the increase of gypsum content can increase the total hydration heat of the system.

As mentioned above, LP can retard the early hydration reaction of C_4_A_3_$\bar{S}$-LP-CaSO_4_·2H_2_O-H_2_O system regardless of gypsum content. The total hydration heat of systems also decreases after the addition of LP. The hydration kinetics of the LP containing system are associated with the content and particle size of LP [37]. When the content of LP is high or the particle size is large, the hydration reaction of system can be retarded.

### 3.2. XRD Analysis

XRD patterns of the C_4_A_3_$\bar{S}$-LP-CaSO_4_·2H_2_O-H_2_O system (M-value = 1) at different curing ages are presented in Figure 4. In the sample of YG1L0, the primary reaction products are ettringite, aluminium hydroxide and AFm. The formation of ettringite is due to the reaction of ye’elimite with gypsum (see Equation (3)). This can be found at all curing ages. When all gypsum is consumed, the remaining ye’elimite reacts with water to form AFm (see Equation (4)). The intensity of the characteristic peak of gypsum (2θ = 11.4°) decreases sharply due to the fast reaction of ye’elimite. At 7 days, it is therefore hard to find the characteristic peak of gypsum. In other words, all gypsum is depleted in this period.
(3)$$C_{4}A_{3}\bar{S}+2C\bar{S}H_{2}+34H\to C_{6}A{\bar{S}}_{3}H_{32}(\mathrm{AFt})+2AH_{3}$$
(4)$$C_{4}A_{3}\bar{S}+18H\to C_{4}A\bar{S}H_{12}(\mathrm{AFm})+2AH_{3}$$

In the samples containing LP, the phases of ettringite and AFm are all found. Equations (3) and (4) all happen with the presence of LP. Besides, the characteristic peak (2θ = 11.6°) of monocarboaluminate (Mc) appears in the LP containing system. The generation of Mc is related to the reaction between ye’elimite and LP (see Equation (5)). It is interesting that hemicarbonate (Hc), another type of calcium carboaluminate, is not found in all samples. This is different from the LP-CSA cement system, in which only Hc is produced when LP participates in the hydration reaction [23]. This is because of the different pH value between the LP-CSA cement system and the C_4_A_3_$\bar{S}$-LP-CaSO_4_·2H_2_O-H_2_O system. It was stated that the pH value of solution can influence the solubility of LP [38]. The solubility of LP decreases with the increase of the pH value. As we know, CSA cement contains alkali metal ions such as sodium and potassium. Its pH value is between 12 and 13 [39], which is higher than that of the C_4_A_3_$\bar{S}$-LP-CaSO_4_·2H_2_O-H_2_O system [40]. Therefore, in the C_4_A_3_$\bar{S}$-LP-CaSO_4_·2H_2_O-H_2_O system, the dissolution of LP is rapid. Accordingly, the dissolved LP can sufficiently react with ye’elimite and produce Mc. The intermediate reaction, viz. producing Hc, is skipped.
(5)$$3C_{4}A_{3}\bar{S}+2C\bar{C}+72H\to 2C_{4}A\bar{C}H_{11}(\mathrm{Mc})+C_{3}A\cdot 3C\bar{S}\cdot H_{32}+6AH_{3}$$

To further discuss the effect of LP on the depletion time of gypsum and the formation time of Mc, detailed XRD patterns from 8° to 13° are presented in Figure 5. The characteristic peaks of gypsum for YG1L0, YG1L5, YG1L15 and YG1L25 samples disappear at 7 days, 3 days, 1 day and 12 h, respectively. Mc forms after the depletion of gypsum for the samples containing LP. It can be observed that the formation time of Mc shortens with the increase of LP dosage. As mentioned above, in the sample containing LP, gypsum primarily reacts with ye’elimite to produce ettringite. Now, it indicates that the reaction of ye’elimite with gypsum is indeed accelerated. This is attributed to the increased effective water to binder ratio due to the presence of LP.

When the M-value is 2, the XRD patterns of the C_4_A_3_$\bar{S}$-LP-CaSO_4_·2H_2_O-H_2_O system at different curing ages are shown in Figure 6. In the LP-free system, the characteristic peak of gypsum still appears at 7 days. As discussed in Section 3.1, C_4_A_3_$\bar{S}$ can fully react with gypsum theoretically to form ettringite when the M-value is 2. From the results of XRD, it can be seen that the reaction between C_4_A_3_$\bar{S}$ and gypsum continues after the curing age of 7 days. For the samples with LP, the characteristic peak of gypsum disappears at 7 days and 12 h for YG2L15 and YG2L25, respectively. This seems like the results when the M-value is 1.

In Figure 6, the reaction products are ettringite and AH_3_ in the sample of YG2L0. When LP is used, the reaction products are still ettringite and AH_3_. It is remarkable that the formation of Mc does not happen in the samples containing LP. In other words, in this system, LP cannot participate in the reaction, which is different from the system with M-value = 1. The reactivity of gypsum is higher than that of LP in the system, therefore, the gypsum reaction with ye’elimite forms ettringite. In addition, ye’elimite can react completely with gypsum when the M-value is 2 (see Equation (3)). Therefore, the LP cannot participate in the reaction when the M-value is 2.

### 3.3. Chemical Shrinkage

CS is the absolute volume change of hydraulic cement paste due to the hydration reaction of pastes. The evolution of CS helps to understand the hydration behavior of cements. The development of the CS for the C_4_A_3_$\bar{S}$-LP-CaSO_4_·2H_2_O-H_2_O system within 72 h is shown in Figure 7. With the M-value of 1 (see Figure 7a), the CS of all samples rapidly increases within 8 h due to the continuous reaction of ye’elimite, and then levels off with increasing the curing age. There are two main differences in these curves resulting from the addition of LP. In the first place, the time when the curve flattens out is shortened due to the addition of LP. The more the LP dosage, the shorter the time. This indicates that the hydration of ye’elimite is accelerated with the incorporation of LP, and this result corresponds to the XRD results. In the next place, the total CS decreases with the increase of the LP dosage. The incorporation of LP can promote the hydration reaction of the system. In addition, LP can reduce the amount of total binder, resulting in a decrease of total CS.

When the M-value is 2 (see Figure 7b), the time when the curve flattens out is also shortened due to the addition of LP. However, it seems that the difference of LP dosage has no obvious influence on this time. As mentioned in the XRD results, the LP does not participate the chemical reaction in the system with an M-value of 2. The shortened flattens out time caused by the addition of LP is related to the nucleation effect. The total CS decreases with the increase of LP dosage, which is due to the addition of LP reducing the amount of total binder.

### 3.4. Thermodynamic Modelling

The thermodynamic modelling method can be used to calculate the phase assemblages and mineral composition of ye’elimite-containing systems at equilibrium condition [41,42]. The final hydration products of the C_4_A_3_$\bar{S}$-LP-CaSO_4_·2H_2_O-H_2_O system with different LP dosages are calculated in this research. Figure 7 shows the results of thermodynamic modelling.

As illustrated in Figure 8a, the hydration products of the LP-free system rely on the content of gypsum. In the system with less gypsum (M-value < 2), the final hydration products are AFm, ettringite and aluminum hydroxide. The amount of ettringite increases with the increase of gypsum content due to the continuous reaction between gypsum and ye’elimite. Meanwhile, the content of AFm phase decreases with the increasing gypsum dosage. Ye’elimite can completely react with gypsum to form ettringite when the M-value is 2. The calculated results from thermodynamic modelling are in accordance with the results of XRD (see Figure 4). With the presence of 5 wt.% LP (see Figure 8b), Mc can form due to the reaction between LP and ye’elimite. However, the formation of Mc does not occur when the M-value increases to 1.26. From the results of Figure 5, it can be seen that there is no Mc formation in the system with the M-value of 2. This means that the reaction between ye’elimite and gypsum is prior to the reaction between ye’elimite and LP. Figure 8c,d presents the thermodynamic modelling results of the systems containing 15 wt.% and 25 wt.% LP, respectively. The content of Mc decreases with the increase of gypsum content, which is similar to the system containing 5 wt.% LP. Furthermore, no AFm is found in these two systems, and unreacted LP is always present in the systems. This indicates that the reaction of LP is limited, which is related to the solubility of LP.

## 4. Conclusions

This study explored the early hydration process of the C_4_A_3_$\bar{S}$-LP-CaSO_4_·2H_2_O-H_2_O systems by combining experimental research and thermodynamic modelling methods. Based on the results and discussions, the following conclusions can be drawn:(1)The systems with 5 wt.% LP have a comparable hydration heat evolution with LP-free systems.(2)LP can retard the formation of the AFm phase, especially when the LP content is 25 wt.%.(3)The formation of Mc is highly related to the M-value of system. When the M-value is 1, LP can participate in the hydration reaction to produce Mc; when the M-value is 2, LP cannot participate in the hydration reaction. Hc is not found in all systems.(4)LP can reduce the total CS of all systems regardless of M-value. The level-off time shortens with the increase of LP dosage, especially when the LP content is 25 wt.%.(5)Mc is no longer generated in all systems when the M-value exceeds 1.27, according to the thermodynamic modelling results. The content of AFm decreases with the increase of the M value and LP dosage. Excess LP always exists in the systems containing 15 wt.% and 25 wt.% LP.

The early hydration behavior of C_4_A_3_$\bar{S}$-LP-CaSO_4_·2H_2_O-H_2_O systems were studied in this paper. According to the results of this study, it can provide theoretical guidance for the reasonable use of LP in sulphoaluminate cement, which is helpful to further reduce carbon emissions and adjust the performance of sulphoaluminate cement.

## Figures and Tables

**Figure 1 materials-15-06645-f001:**
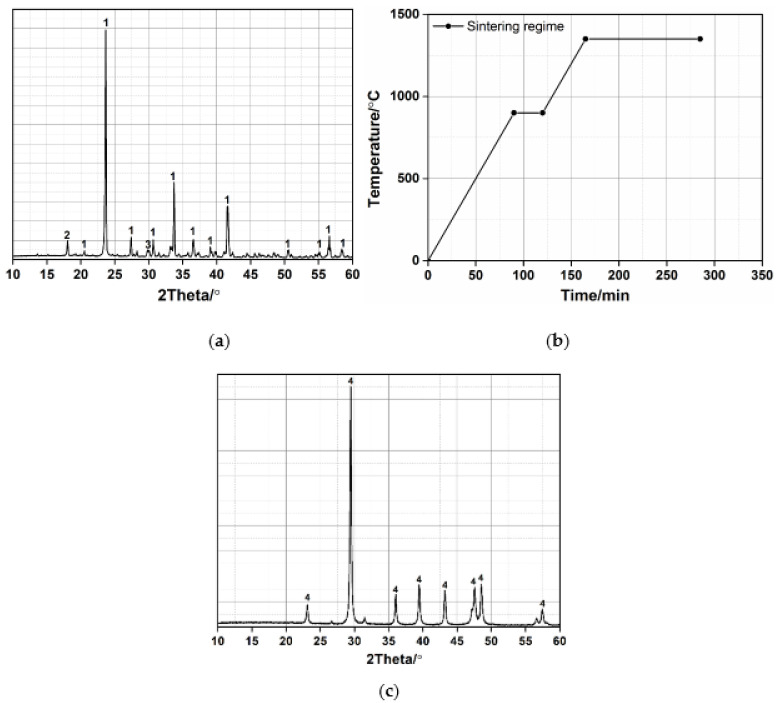
XRD patterns of synthetic ye’elimite (**a**); d sintering regime of ye’elimite (**b**); and limestone powder (**c**) (1-ye’elimite; 2—mayenite; 3—calcium aluminate; 4—calcite).

**Figure 2 materials-15-06645-f002:**
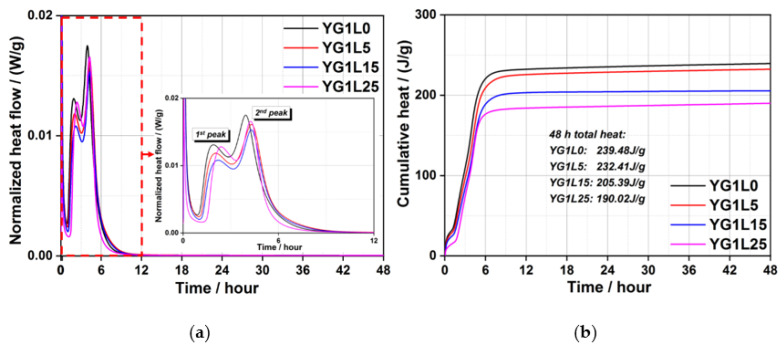
Hydration heat curves (**a**); and cumulative heat curves (**b**) of the C_4_A_3_$\bar{S}$-LP-CaSO_4_·2H_2_O-H_2_O system (M-value = 1).

**Figure 3 materials-15-06645-f003:**
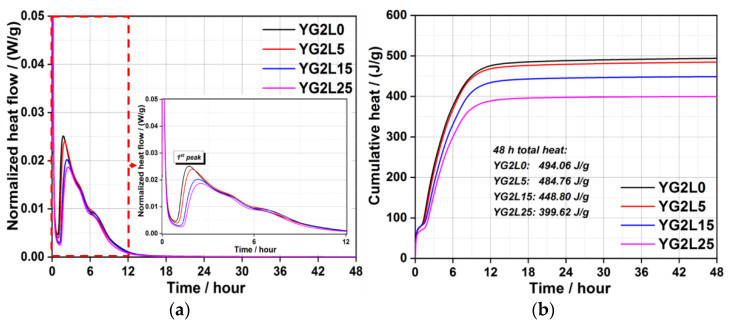
Hydration heat flow curves (**a**); and cumulative heat curves (**b**) of the C_4_A_3_$\bar{S}$-LP-CaSO_4_·2H_2_O-H_2_O system (M-value = 2).

**Figure 4 materials-15-06645-f004:**
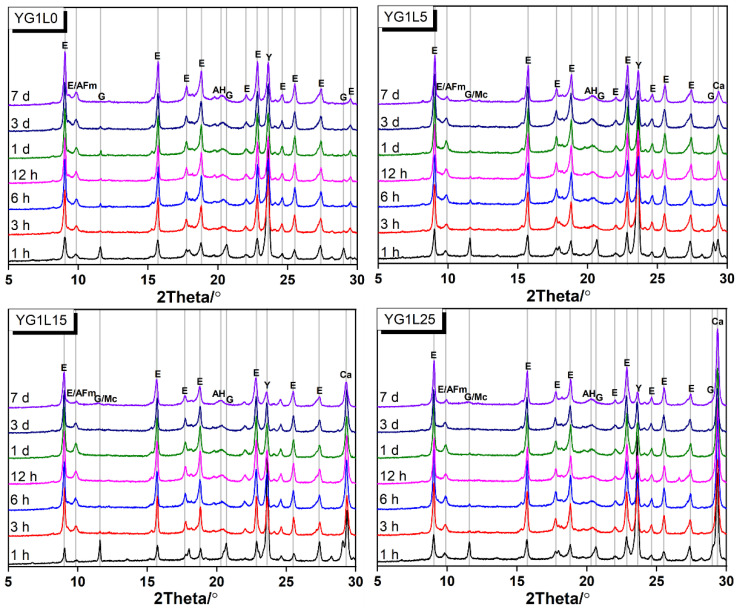
XRD patterns of the C_4_A_3_$\bar{S}$-LP-CaSO_4_·2H_2_O-H_2_O system (M-value = 1) (E-ettringite, G-gypsum, Mc-monocarboaluminate, AH—aluminium hydroxide, Ca—calcite).

**Figure 5 materials-15-06645-f005:**
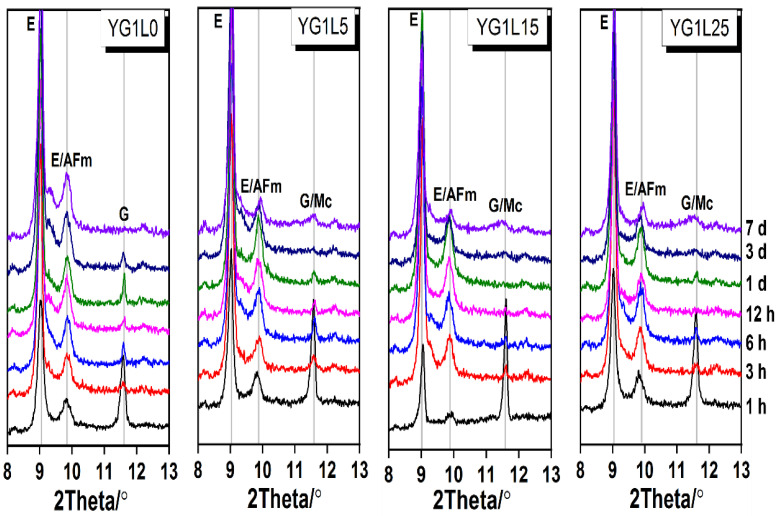
XRD patterns at low diffraction angle (8–13°) for the C_4_A_3_$\bar{S}$-LP-CaSO_4_·2H_2_O-H_2_O system (M-value = 1) (E-ettringite, G—gypsum, Mc—monocarboaluminate).

**Figure 6 materials-15-06645-f006:**
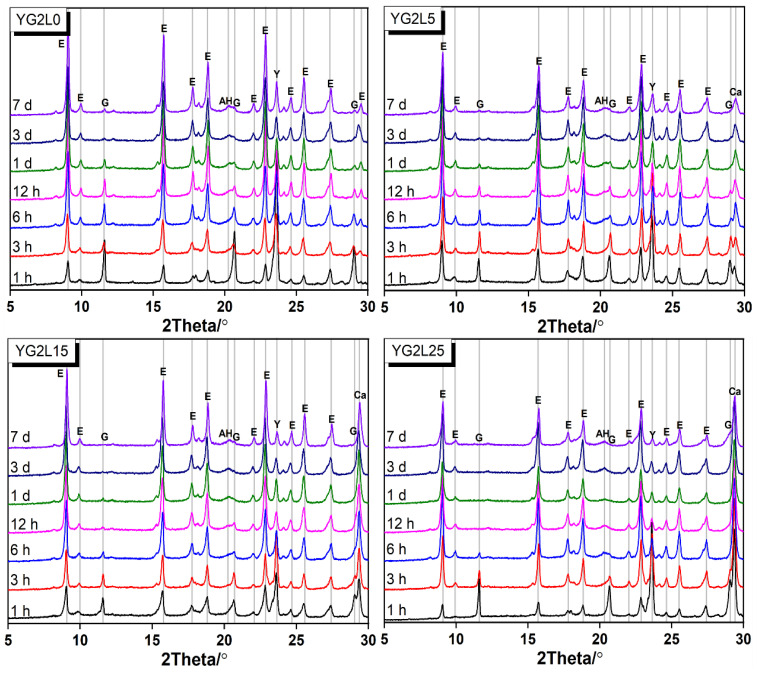
XRD patterns of the C_4_A_3_$\bar{S}$-LP-CaSO_4_·2H_2_O-H_2_O system (M-value = 2) (E—ettringite, G—gypsum, AH—aluminium hydroxide, Ca—calcite).

**Figure 7 materials-15-06645-f007:**
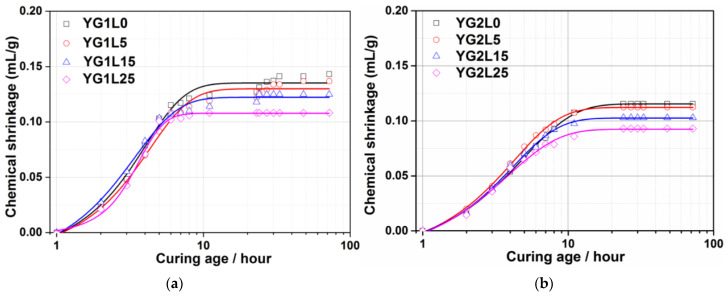
CS of the C_4_A_3_$\bar{S}$-LP-CaSO_4_·2H_2_O-H_2_O systems. (**a**) M-value = 1; (**b**) M-value = 2.

**Figure 8 materials-15-06645-f008:**
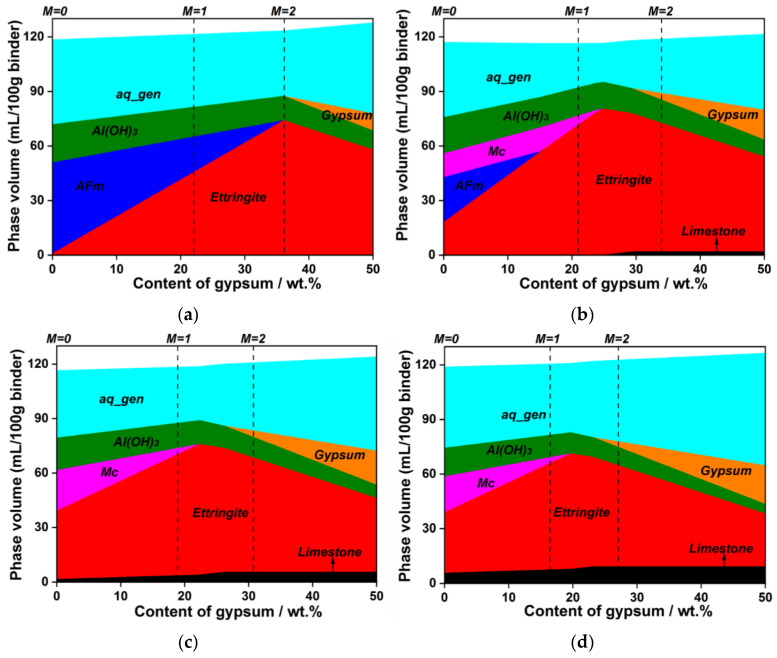
Thermodynamic modelling results of C_4_A_3_$\bar{S}$-LP-CaSO_4_·2H_2_O-H_2_O system. (**a**) LP = 0; (**b**) LP = 5 wt.%; (**c**) LP = 15 wt.%; (**d**) LP = 25 wt.%.

**Table 1 materials-15-06645-t001:** Chemical composition of LP.

Oxides	CaO	SiO_2_	Al_2_O_3_	SO_3_	Fe_2_O_3_	MgO	K_2_O	Na_2_O	LOI
wt.%	55.34	0.68	0.09	0.017	0.09	0.22	0.004	0.002	43.5

**Table 2 materials-15-06645-t002:** The mixture design used in this study.

Number	M-Value	Water/Binder	Proportion/wt.%		
			$C_{4}A_{3}\bar{S}$	CaSO_4_·2H_2_O	Limestone Powder
YG1L0	1	1	78.0	22.0	0
YG1L5			74.1	20.9	5.0
YG1L15			66.3	18.7	15.0
YG1L25			58.5	16.5	25.0
YG2L0	2		63.9	36.1	0
YG2L5			60.7	34.3	5.0
YG2L15			54.3	30.7	15.0
YG2L25			47.9	27.1	25.0

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
