# Peer review of "Effects of Limestone Powder on the Early Hydration Behavior of Ye’elimite: Experimental Research and Thermodynamic Modelling"

_materials, 2022, doi:10.3390/ma15196645_

Round 1
Reviewer 1 Report
The article is about experimental research and thermodynamic modelling and the effects of limestone powder on the early hydration behavior of Ye’elimite. However, some issues must to be addressed:
- Abstract: Please start by expressing the aim of this paper, followed by the rest of the information. Also, please define or try to avoid using abbreviations in the abstract. Typically, the abstract should provide a broad overview of the entire project, summarize the results, and present the implications of the research or what it adds to its field.
- The introduction section must to be improved by adding in references section the newest articles related to the topic from 2020-2022.
- The bibliographic foundation is important and well executed, however some new discussions should be inserted, authors should consider some works in the literature, such as: DOI 10.1088/1757-899X/374/1/012019.
- Table 1: how was determined the chemical composition? Fe2O3 … why not Fe3O4 ?????!!!!
- The results are merely presented, not properly discussed. Please add explanations for the observed changes. Please give an extended discussion on the obtained results and correlate your findings with previous literature studies and prospective applications.
- More analysis and interpretation of the results should be added for a clearer understanding of observed experimental phenomena.
- The authors must to provide some details about importance of the research and their applicability.
- Please rewrite the conclusions in a more quantitative form and enhance the clarity of the conclusion section in order to highlight the results obtained.
- General check-up and correction of the English language is suggested. There are still some minor typos and grammatical errors.
The author needs to address the abovementioned points for the betterment of the manuscript.
Author Response
Dear editor,
Thanks for your help regarding the handling of the paper materials-1902252 for Materials. We thank you and the reviewers for giving us constructive suggestions again which helps us in depth to improve the quality of the paper. Based on the comments and suggestions, we submitted a revised version of this manuscript. We also marked revised part in blue in the revised manuscript.
Thank you very much for your help.
Sincerely yours
Jian Ma
20-Sep-2022
Reviewers’ comments:
Reviewer #1:
The article is about experimental research and thermodynamic modelling and the effects of limestone powder on the early hydration behavior of Ye’elimite. However, some issues must to be addressed:
- Abstract: Please start by expressing the aim of this paper, followed by the rest of the information. Also, please define or try to avoid using abbreviations in the abstract. Typically, the abstract should provide a broad overview of the entire project, summarize the results, and present the implications of the research or what it adds to its field.
Re: Thanks for your suggestion. In the revised manuscript, this part is revised as follows:
Line 14-28:
In order to explore the effects of limestone powder and gypsum on the early hydration of ye’elimite, the hydration behavior of C4A3SÌ…-LP-CaSO4·2H2O-H2O systems are researched. The hydration behavior of systems are researched by employing isothermal calorimetry, XRD technique and chemical shrinkage. Thermodynamic modelling method is used to predict the equilibrium phase assemblages. The results show that the system with 5 wt.% LP has a comparable hydration heat evolution to limestone powder free systems. Limestone powder can participate in the hydration reaction to produce monocarboaluminate in the system with M-value (molar ratio of gypsum to ye’elimite) of 1, but monocarboaluminate is not found in the system with M-value of 2. The level off time of chemical shrinkage shortens with the increase of limestone powder dosage. Thermodynamic modelling results show that monocarboaluminate is no longer formed in all systems when M-value exceed 1.27, which is correspond to the XRD results. This study can provide theoretical guidance for the rational utilization of limestone powder in calcium sulphoaluminate cement.
- The introduction section must to be improved by adding in references section the newest articles related to the topic from 2020-2022.
Re: Thanks for your suggestion. The relevant literatures have been cited in the revised manuscript.
Line 381-382:
[2] J.S.J. van Deventer, C.E.White, R. J. Myers. A Roadmap for Production of Cement and Concrete with Low‑CO2 Emissions, Waste Biomass Valori., 12 (2020) 4745-4775.
Line 393-394:
[8] H.N.Yoon, J.Seo, S.Kim, et al, Hydration of calcium sulfoaluminate cement blended with blast-furnace slag, Constr. Build. Mater., 268 (2021) 121214.
Line 401-403:
[12] J.Ma, H.N.Wang, Z.Q.Yu, et al.A systematic review on durability of calcium sulphoaluminate cement-based materials in chloride environment, J Sustain Cem-Based, (2022) 1-12.
- The bibliographic foundation is important and well executed, however some new discussions should be inserted, authors should consider some works in the literature, such as: DOI 10.1088/1757-899X/374/1/012019.
Re: Thanks for your suggestion. The literature has been inserted in this article. In the revised manuscript, this part is revised as follows:
Line 390-392:
[7] D.D.Burduhos Nergis, M.M.A.B. Abdullah, P. Vizureanu, et al. Geopolymers and Their Uses: Review, IOP Conference Series: Materials Science and Engineering, 374 (2018) 012019
- Table 1: how was determined the chemical composition? Fe2O3 … why not Fe3O4 ?
Re: The chemical composition of material is determined by X-ray fluorescence. Only Fe2O3 is detected according to the result of X-ray fluorescence.
- The results are merely presented, not properly discussed. Please add explanations for the observed changes. Please give an extended discussion on the obtained results and correlate your findings with previous literature studies and prospective applications.
Re: Thanks for your suggestion. In the revised manuscript, this part is revised as follows:
Line 202-207:
As mentioned above, LP can retard the early hydration reaction of C4A3SÌ…-LP-CaSO4·2H2O-H2O system regardless of gypsum content. The total hydration heat of systems also decreases after addition of LP. It was reported that the hydration kinetics of LP containing cement system is related to the content and particle size of LP [37]. When the content of LP is high or the particle size is large, the hydration reaction of system can be retarded.
Line 470-472:
[37] D.H.Wang, C.J. Shi, N.Farzadnia, et al., A review on use of limestone powder in cement-based materials: Mechanism, hydration and microstructures, Constr. Build. Mater., 181 (2018) 659-672.
- More analysis and interpretation of the results should be added for a clearer understanding of observed experimental phenomena.
Re: Thanks for your suggestion. In the revised manuscript, this part is revised as follows:
Line 265-269:
The reactivity of gypsum is higher than that of LP in the system, therefore, the gypsum reaction with ye’elimite to form ettringite. In addition, ye’elimite can react completely with gypsum when M-value is 2 (see Eq.3). Therefore, the LP cannot participate in the reaction when the M-value is 2.
- The authors must to provide some details about importance of the research and their applicability.
Re: Thanks for your suggestion. In the revised manuscript, this part is revised as follows:
Line 351-354:
The early hydration behavior of C4A3SÌ…-LP-CaSO4·2H2O-H2O systems were studied in this paper. According to the results of this study, it can provides theoretical guidance for rational utilization of LP in sulphoaluminate cement, which is helpful to further reduce carbon emissions and adjust performance of sulphoaluminate cement.
- Please rewrite the conclusions in a more quantitative form and enhance the clarity of the conclusion section in order to highlight the results obtained.
Re: Thanks for your suggestion. In the revised manuscript, this part is revised as follows:
Line 336-350
1) The systems with 5 wt.% LP have a comparable hydration heat evolution with LP free systems.
2) LP can retard the formation of AFm phase, especially when the LP content is 25 wt.%.
3) The formation of Mc is highly related to the M-value of system. When the M-value is 1, LP can participate in the hydration reaction to produce Mc; when the M-value is 2, LP cannot participate in the hydration reaction in the system with M-value of 2. Hc is not found in all systems.
4) LP can decrease the total chemical shrinkage of all systems regardless of M-value. The level off time shortens with the increase of LP dosage, especially when the LP content is 25 wt.%.
5) Mc is no longer generated in all systems when M-value exceed 1.27 according to the thermodynamic modelling results. The content of AFm decreases with the increase of M value and LP dosage. Excess LP always exist in the systems containing 15 wt.% and 25 wt.% LP.
- General check-up and correction of the English language is suggested. There are still some minor typos and grammatical errors.
Re: Thanks for your suggestion. This manuscript was checked carefully, including English writing and English usage and revised in blue.

Reviewer 2 Report
- The paper is interesting and relatively scientifically novel.
- The abstract covers all the mall points provided in the research. Abbreviations should be removed from the abstract.
- Key words should not repeat the terms already used in the title.
- The text is relatively readable; however, the authors have to provide some effort and rearrange text to sound more scientific. The text should be corrected by native English language speaker. Also, all typing and spelling mistakes should be removed, and all terms should be checked.
- Introduction – line 32: “The cement industry currently 32 accounts for 5% of global carbon dioxide production”. The number is higher than 5%.
- Introduction – line 41: “Ye’elimite (C4A3SÌ…) is the main mineral phase in CSA cement, which is about 50- 80 wt.%” 50-80% of what?
- Line 54: “In addition to being raw material to generate Portland cement, limestone powder 54 (LP) can also be used to partial replace cement to reduce the carbon emission” How? Please do elaborate.
- References are slightly out-dated. There are no new references (from year 2020-2022).
- Also, the research under following DOI Number https://doi.org/10.1016/j.cemconres.2019.03.024 has been omitted from the references. This research is pretty similar to the one presented here. It is peculiar that this work is omitted especially since it appears as the first result in the search.
- Materials and methods: how synthetic ye’elimite will make cement production more sustainable? Or economical? Or how it will save energy if sintering at 1350C is included?
- Fig 1: XRD of Ye’elimite should be explained.
- Fig 1b – sintering regime is not necessary. Sintering regime can be explain in the text.
- Methods: why DTA analysis has not been included?
- Results are well explained.
- Conclusions are logical and derived from the discussion presented before.
Author Response
Dear editor,
Thanks for your help regarding the handling of the paper materials-1902252 for Materials. We thank you and the reviewers for giving us constructive suggestions again which helps us in depth to improve the quality of the paper. Based on the comments and suggestions, we submitted a revised version of this manuscript. We also marked revised part in blue in the revised manuscript.
Thank you very much for your help.
Sincerely yours
Jian Ma
20-Sep-2022
Reviewers’ comments:
Reviewer #2:
The paper is interesting and relatively scientifically novel.
- The abstract covers all the mall points provided in the research. Abbreviations should be removed from the abstract.
Re: Thanks for your suggestion. In the revised manuscript, this part is revised as follows:
Line 14-28
In order to explore the effects of limestone powder and gypsum on the early hydration of ye’elimite, the hydration behavior of C4A3SÌ…-LP-CaSO4·2H2O-H2O systems are researched. The hydration behavior of systems are researched by employing isothermal calorimetry, XRD technique and chemical shrinkage. Thermodynamic modelling method is used to predict the equilibrium phase assemblages. The results show that the system with 5 wt.% LP has a comparable hydration heat evolution to limestone powder free systems. Limestone powder can participate in the hydration reaction to produce monocarboaluminate in the system with M-value (molar ratio of gypsum to ye’elimite) of 1, but monocarboaluminate is not found in the system with M-value of 2. The level off time of chemical shrinkage shortens with the increase of limestone powder dosage. Thermodynamic modelling results show that monocarboaluminate is no longer formed in all systems when M-value exceed 1.27, which is correspond to the XRD results. This study can provide theoretical guidance for the rational utilization of limestone powder in calcium sulphoaluminate cement.
- Key words should not repeat the terms already used in the title.
Re: Thanks for your suggestion. In the revised manuscript, this part is revised as follows:
Line 31-32:
Hydration, limestone powder content, gypsum content, thermodynamic modelling, chemical shrinkage
- The text is relatively readable; however, the authors have to provide some effort and rearrange text to sound more scientific. The text should be corrected by native English language speaker. Also, all typing and spelling mistakes should be removed, and all terms should be checked.
Re: Thanks for your suggestion. This manuscript was checked carefully, including English writing and English usage and revised in blue.
- Introduction – line 32: “The cement industry currently 32 accounts for 5% of global carbon dioxide production”. The number is higher than 5%.
Re: Thanks for your careful work. In the revised manuscript, this part is revised as follows:
Line 35-36:
The cement industry currently accounts for 5-8% of global carbon dioxide production [2]
[2] J.S.J. van Deventer, C.E.White, R. J. Myers. A Roadmap for Production of Cement and Concrete with Low‑CO2 Emissions, Waste and Biomass Valori., 12 (9) 4745-4775.
- Introduction – line 41: “Ye’elimite (C4A3SÌ…) is the main mineral phase in CSA cement, which is about 50- 80 wt.%” 50-80% of what?
Re: Thanks for your careful work. In the revised manuscript, this part is revised as follows:
Line 44-45:
Ye’elimite (C4A3SÌ…) is the main mineral phase in CSA cement, which is about 50-80 wt.% of CSA cement [13].
- Line 54: “In addition to being raw material to generate Portland cement, limestone powder 54 (LP) can also be used to partial replace cement to reduce the carbon emission” How? Please do elaborate.
Re: The decomposition of raw materials such as limestone are the major sources of carbon emissions. Generally, 0.87 t of CO2 can be produced in the production of 1 t of Portland cement clinker. Limestone powder can be used as supplementary cementitious material to partial replace cement clinker, thereby reducing the consumption of Portland cement clinker and indirectly reducing carbon emissions.
- References are slightly out-dated. There are no new references (from year 2020-2022).
Re: Thanks for your suggestion. The relevant literatures have been cited in the revised manuscript.
Line 381-382:
[2] J.S.J. van Deventer, C.E.White, R. J. Myers. A Roadmap for Production of Cement and Concrete with Low‑CO2 Emissions, Waste Biomass Valori., 12 (2020) 4745-4775.
Line 393-394:
[8] H.N.Yoon, J.Seo, S.Kim, et al, Hydration of calcium sulfoaluminate cement blended with blast-furnace slag, Constr. Build. Mater., 268 (2021) 121214.
Line 401-403:
[12] J.Ma, H.N.Wang, Z.Q.Yu, et al.A systematic review on durability of calcium sulphoaluminate cement-based materials in chloride environment, J Sustain Cem-Based, (2022) 1-12.
- Also, the research under following DOI Number
https://doi.org/10.1016/j.cemconres.2019.03.024 has been omitted from the references. This research is pretty similar to the one presented here. It is peculiar that this work is omitted especially since it appears as the first result in the search.
Re: Thanks for your carful work. Zajac et al. (DOI: https://doi.org/10.1016/j.cemconres.2019.03.024) mainly researched ye’elimite containing system and focused on thermodynamic modelling. In this research, the effect of LP on the hydration of the C4A3SÌ…-LP-CaSO4·2H2O-H2O systems were studied by combining experiment and thermodynamic simulation, which is different from the work done by Zajac et al. And this reference has been cited in section 3.4. In the revised manuscript, this part is revised as follows:
Line 483-484:
[42] M.Zajac, J. Skocek, F Bullerjahn, et al. Early hydration of ye’elimite: Insght from thermodynamic modelling, 120 (2019) 152-163.
- Materials and methods: how synthetic ye’elimite will make cement production more sustainable? Or economical? Or how it will save energy if sintering at 1350C is included?
Re: The calcination temperature for producing tradition Portland cement clinker is 1450ËšC, while the temperature for ye’elimite is 1350ËšC, thus reducing carbon emissions.
- Fig 1: XRD of Ye’elimite should be explained.
Re: Thanks for your suggestion. In the revised manuscript, this part is revised as follows:
Line 88-89:
It can be seen that the main mineral composition of ye’elimite phase is ye’elimite, mayenite and calcium aluminate.
- Fig 1b – sintering regime is not necessary. Sintering regime can be explain in the text.
Re: Thanks for your suggestion. It is more intuitive to use sintering regime image, therefore, it is necessary to retain the sintering regime image
- Methods: why DTA analysis has not been included?
Re: Thanks for your suggestion. We will use DTA analysis to further research the hydration of this system in our next work.
- Results are well explained.
Re: Thanks for your work.
- Conclusions are logical and derived from the discussion presented before.
Re: Thanks for your work.

Reviewer 3 Report
The manuscript entitled "Effects of Limestone Powder on the Early Hydration Behavior of Ye’elimite: Experimental Research and Thermodynamic Modelling." presents an interesting experimental study conducted on the effect of limestone on the hydration behavior of synthetic ye’elimite. However, the paper has multiple issues that must be addressed. The paper needs minor revisions before it is processed further, some comments follow:
Methods and Materials Section
XRD spectra of ye’elimite: Why some peaks were considered instead of others. As can be seen in the spectra, some clear peaks around 21, 40, 50, 55, and 58 were not evaluated. Please evaluate all the peaks (or at least those with significant intensity) and make corresponding appreciations. Also, a quantitative evaluation of phases could enhance the quality of the research.
LP analysis: Please provide the XRD analysis of all raw materials, otherwise, the XRD of the synthesized samples will be hard to read/understand (which peaks come from the raw material and which peaks correspond to the synthesis reaction). As can be seen from the chemical composition some phases containing Fe should be detected (!!!!!if hematite is detected please keep Fe2O3, if magnetite is detected please replace the Fe oxide with Fe3O4, if both are detected please present it as FexOy).
What was the rationale for choosing these mixtures? Why the authors didn’t design the sample considering” the Taguchi method? and what was the rationale for choosing the C4A3SÌ… to CaSO4·2H2O ratio?
Test methods
“The XRD data were recorded from 5º to 30º at” why only this range? Please make corresponding comments in the manuscript.
Results and discussions
XRD analysis of samples – same comments as for Figure 1. Please identify all phases.
Raw materials- Overall comment.
What was the rationale in making some experiments/determinations for some raw materials and others for other raw materials, i.e., the XRD was analyzed for ye’eliomite, while for LP only the chemical composition was studied? Please provide the results for the same analyzing technique to show a clear comparison between these materials...
Author Response
Dear editor,
Thanks for your help regarding the handling of the paper materials-1902252 for Materials. We thank you and the reviewers for giving us constructive suggestions again which helps us in depth to improve the quality of the paper. Based on the comments and suggestions, we submitted a revised version of this manuscript. We also marked revised part in blue in the revised manuscript.
Thank you very much for your help.
Sincerely yours
Jian Ma
20-Sep-2022
Reviewers’ comments:
Reviewer #3:
The manuscript entitled "Effects of Limestone Powder on the Early Hydration Behavior of Ye’elimite: Experimental Research and Thermodynamic Modelling." presents an interesting experimental study conducted on the effect of limestone on the hydration behavior of synthetic ye’elimite. However, the paper has multiple issues that must be addressed. The paper needs minor revisions before it is processed further, some comments follow:
- XRD spectra of ye’elimite: Why some peaks were considered instead of others. As can be seen in the spectra, some clear peaks around 21, 40, 50, 55, and 58 were not evaluated. Please evaluate all the peaks (or at least those with significant intensity) and make corresponding appreciations. Also, a quantitative evaluation of phases could enhance the quality of the research.
Re: Thanks for your suggestion. All peaks of XRD spectra of ye’elimite has been marked. In the revised manuscript, this part is revised as follows:
Line 94:
(a) (b)
- LP analysis: Please provide the XRD analysis of all raw materials, otherwise, the XRD of the synthesized samples will be hard to read/understand (which peaks come from the raw material and which peaks correspond to the synthesis reaction). As can be seen from the chemical composition some phases containing Fe should be detected (!!!!!if hematite is detected please keep Fe2O3, if magnetite is detected please replace the Fe oxide with Fe3O4, if both are detected please present it as FexOy).
Re: Thanks for your suggestion. The XRD analysis of LP has been added in manuscript. The content of phases containing Fe are very low, and can not detected by XRD. According to the X-ray fluorescence analysis, only Fe2O3 is detected. In the revised manuscript, this part is revised as follows:
Line 96-100:
(c)
Fig. 1 XRD patterns of synthetic ye’elimite (a), d sintering regime of ye’elimite (b) and limestone powder (c)
(1-ye’elimite; 2-mayenite; 3-calcium aluminate; 4-calcite)
- What was the rationale for choosing these mixtures? Why the authors didn’t design the sample considering” the Taguchi method? and what was the rationale for choosing the C4A3SÌ… to CaSO4·2H2O ratio?
Re: The molar ratio of gypsum to ye’elimite (M-value) determines the type and content of the hydration products. When the M-value is 2, ye’elimite reacts with gypsum to form AFt and aluminum hydroxide (see Eq. (1)). If the M-value is 1, both AFt and AFm phase exist in the hydration products (see Eq. (2)). Therefore, these mixtures were selected by this research to systematically research the C4A3SÌ…-LP-CaSO4·2H2O-H2O system.
|
(1) |
|
|
(2) |
- “The XRD data were recorded from 5º to 30º at” why only this range? Please make corresponding comments in the manuscript.
Re: The characteristic diffraction peaks of the main hydration products of C4A3SÌ…-LP-CaSO4·2H2O-H2O system mainly appears in the range of 5º -30º, therefore, the XRD data were recorded from 5º to 30º in this research. In the revised manuscript, this part is revised as follows:
Line 126-129:
The characteristic diffraction peaks of the main hydration products of C4A3SÌ…-LP-CaSO4·2H2O-H2O system mainly appears in the range of 5º -30º in XRD pattern, therefore, the XRD data were recorded from 5º to 30º at the scan speed of 5º/min.
- XRD analysis of samples-same comments as for Figure 1. Please identify all phases.
Re: Thanks for your suggestion. In the revised manuscript, this part is revised as follows:
Line 223-236:
Fig. 4 XRD patterns of the C4A3SÌ…-LP-CaSO4·2H2O-H2O system (M-value=1)
(E-ettringite, G-gypsum, Mc-monocarboaluminate, AH- aluminium hydroxide, Ca-calcite)
Line 270-272:
Fig. 6 XRD patterns of the C4A3SÌ…-LP-CaSO4·2H2O-H2O system (M-value=2)
(E-ettringite, G-gypsum, AH- aluminium hydroxide, Ca-calcite)
- What was the rationale in making some experiments/determinations for some raw materials and others for other raw materials, i.e., the XRD was analyzed for ye’elimite, while for LP only the chemical composition was studied? Please provide the results for the same analyzing technique to show a clear comparison between these materials...
Re: Thanks for your suggestion. The XRD pattern of LP has been added in the revised manuscript. In the revised manuscript, this part is revised as follows:
Line 94-100:
(a) (b)
(c)
Fig. 1 XRD patterns of synthetic ye’elimite (a), d sintering regime of ye’elimite (b) and limestone powder (c)
(1-ye’elimite; 2-mayenite; 3-calcium aluminate; 4-calcite)

Round 2
Reviewer 1 Report
The article is suitable for publication.
Reviewer 2 Report
The authors corrected the manuscript. It can be accepted for the publication in this form.